# Influence of the Degree of Deacetylation of Chitosan and BMP-2 Concentration on Biocompatibility and Osteogenic Properties of BMP-2/PLA Granule-Loaded Chitosan/β-Glycerophosphate Hydrogels

**DOI:** 10.3390/molecules26020261

**Published:** 2021-01-07

**Authors:** Andrey Vyacheslavovich Vasilyev, Valeriya Sergeevna Kuznetsova, Tatyana Borisovna Bukharova, Timofei Evgenevich Grigoriev, Yuriy Dmitrievich Zagoskin, Irina Alekseevna Nedorubova, Igor Ivanovich Babichenko, Sergey Nicolaevich Chvalun, Dmitry Vadimovich Goldstein, Anatoliy Alekseevich Kulakov

**Affiliations:** 1Research Centre for Medical Genetics, Moskvorechye st., 1, 115478 Moscow, Russia; tilia7@ya.ru (V.S.K.); bukharova-rmt@yandex.ru (T.B.B.); irina0140@gmail.com (I.A.N.); dv@rm7.ru (D.V.G.); kulakov@cniis.ru (A.A.K.); 2Central Research Institute of Dental and Maxillofacial Surgery, Timur Frunze st., 16, 119021 Moscow, Russia; babichenko-ii@rudn.ru; 3Institute of Medicine, Peoples’ Friendship University of Russia (RUDN University), 6 Miklukho-Maklaya st., 117198 Moscow, Russia; 4NRC “Kurchatov Institute”, 1, Akademika Kurchatova pl., 123182 Moscow, Russia; timgrigo@yandex.ru (T.E.G.); zagos@inbox.ru (Y.D.Z.); chvalun_SN@nrcki.ru (S.N.C.); 5Moscow Institute of Physics and Technology (National Research University), 9 Institutskiy per., Dolgoprudny, 141701 Moscow, Russia

**Keywords:** chitosan, deacetylation degree, osteoplastic material, BMP-2, porous polylactide granules, osteoinduction

## Abstract

Compositions based on chitosan/β-glycerophosphate hydrogels with highly porous polylactide granules can be used to obtain moldable bone graft materials that have osteoinductive and osteoconductive properties. To eliminate the influence of such characteristics as chain length, degree of purification, and molecular weight on a designed material, the one-stock chitosan sample was reacetylated to degrees of deacetylation (DD%) of 19.5, 39, 49, 55, and 56. A study of the chitosan/β-glycerophosphate hydrogel with chitosan of a reduced DD% showed that a low degree of deacetylation increased the MSCs (multipotent stromal cells) viability rate in vitro and reduced the leukocyte infiltration in subcutaneous implantation to Wistar rats in vivo. The addition of 12 wt% polylactide granules resulted in optimal composite mechanical and moldable properties, and increased the modulus of elasticity of the hydrogel-based material by approximately 100 times. Excessive filling of the material with PLA (polylactide) granules (more than 20%) led to material destruction at a ~10% strain. Osteoinductive and osteoconductive properties of the chitosan hydrogel-based material with reacetylated chitosan (39 DD%) and highly porous polylactide granules impregnated with BMP-2 (bone morphogenetic protein-2) have been demonstrated in models of orthotopic and ectopic bone formation. When implanted into a critical-size calvarial defect in rats, the optimal concentration of BMP-2 was 10 μg/mL: bone tissue areas filled the entire material’s thickness. Implantation of the material with 50 μg/mL BMP-2 was accompanied with excessive growth of bone tissue and material displacement beyond the defect. Significant osteoinductive and osteoconductive properties of the material with 10 μg/mL of BMP-2 were also shown in subcutaneous implantation.

## 1. Introduction

Annually the number of bone grafting procedures in Europe and America grows by an average of 13–14.4%, and approximately 2.2 million operations are performed every year [1,2,3]. Bone substitutes are mostly used in traumatology, orthopedics, dentistry, and maxillofacial surgery [4]. The existing materials are not ideal because they do not thoroughly combine such qualities as osteoinduction, osteoconduction, biocompatibility, resorption ability, figurability, and ease-of-use [5,6].

The consistency of bone substitutes is divided into bone granules, cement, paste, sponge, and putty. Putty is the most useful material because it can be molded into any shape. However, unlike cements, it cannot cure in the wound and hold its shape [7]. Unlike bone granules, registered putty has a continuous surface that prevents the vessel’s ingrowth and resorption of material, which limit bone formation inside the putty [8]. In this regard, the design of thermosetting biocompatible and bioresorbable putty, with pores for vascular ingrowth and cell migration, is a crucial task.

A possible way is to use a chitosan hydrogel as a base [6]. Chitosan forms thermotropic hydrogel with β-glycerophosphate. After insertion into the wound, it retains a predetermined shape and can prolong the release of osteoinductors. The use of β-glycerophosphate as a crosslinking agent is preferred because the resulting materials do not contain organic solvents or toxic initiators [9]. Moreover, β-glycerophosphate is a component of osteogenic media and promotes osteogenic cell differentiation [10].

The biological properties of the material for bone regeneration are primarily associated with osteoinduction—the ability to stimulate bone tissue formation. It is possible to improve the osteoinductive properties by using BMP-2—the most efficient osteogenic growth factor [11]. We have previously shown that highly porous polylactide granules can be used as a BMP-2 carrier. They can release BMP-2 within six days. Small doses of BMP-2 induce osteogenesis due to prolonged release. Prolonged release should help reduce the risks that may be associated with excessive BMP-2 concentration: hyperostosis, inflammation, and anaplasia [12,13]. The use of PLA granules in combination with a chitosan–glycerophosphate hydrogel will improve the resorption of chitosan by proton acceptance during the resorption of polylactides. Additionally, the granules will improve the mechanical properties of the resulting composition [14].

This study aimed to design a biocompatible, osteoinductive, and easy-to-use material based on chitosan gel and highly porous polylactide granules with BMP-2 for bone tissue regeneration.

## 2. Results

### 2.1. Selection of Biocompatible Components

The effect of the degree of deacetylation (DD%) on chitosan-based hydrogels’ biocompatibility in vivo and in vitro was determined (Figure 1). The reacetylation reaction prepared the chitosan samples with a 19.5–65% degree of deacetylation. The one-stock chitosan sample used as a substrate eliminated differences in impurities, molecular weight variations, and other factors that influence polymer biological properties. A sample with a 19.5% degree of deacetylation was insoluble and could not form gels or gel-like structures. Therefore, it was excluded from further study.

An in vitro MTT test showed that chitosan with a 39 DD% had no cytotoxic effect on MSC (multipotent stromal cells, Figure 1A,B). A higher degree of deacetylation of 49% had a moderate cytotoxic effect. A DD% above 55 resulted in an increased number of dead cells (Figure 1A,B).

The in vivo study in the subcutaneous implantation model showed similar results. A decrease in the degree of deacetylation correlated with a decrease in leukocyte infiltration (*p* = 0.03) (Figure 1C,D). Rare leukocytes were seen by field of view in the implantation site of chitosan with a 39 DD% (Figure 1C,D).

The incorporation of 10–16 wt% highly porous polylactide granules into the chitosan–glycerophosphate hydrogel resulted in a significant decrease in leukocyte infiltration (Figure 1C). At 14 days after subcutaneous implantation, there were 0–2 leukocytes per field of view (Figure 1D). However, numerous foreign-body giant cells and small number of lymphocytes around the polylactide granules were found. This characteristic pattern is typical for polylactide resorption.

### 2.2. Physical Properties of Chitosan-Based Materials

Studying the physico-mechanical properties of hydrogels based on chitosan/β-glycerophosphate showed that the enhancement of the mechanical strength is attributed to the PLA granules (Figure 2). The addition of 12 wt% PLA granules increased the modulus of elasticity approximately 100 times more compared with an unfilled system. The most important factor influencing the elastic modulus in such systems was the overlap of the particles rather than the properties of the gel. So decreasing the degree of deacetylation of the chitosan to 39% did not significantly change the mechanical properties of the composite.

Excessive filling of hydrogel with PLA granules of more than 20% led to material destruction at a ~10% strain (Figure 2). This also caused inconvenience when modeling different shapes in hands. Therefore, a composition containing 12 wt% granules was selected as the base for the material.

### 2.3. Induction of Orthotopic Bone Formation

The biological properties of the hydrogel-based materials of reacetylated chitosan and β-glycerophosphate and highly porous polylactide granules were studied. For this purpose, materials with different concentrations of BMP-2: 0 (without), 10, and 50 μg/mL were implanted into critical-size calvarial bone defects in rats. A critical-size defect is a defect that will not completely heal spontaneously within the lifetime of the animal or the experiment [15]. In this regard, this model was supposed to clearly show the osteoconductive (vascular ingrowth and cell migration) and osteoinductive properties (bone formation in situ) of the material.

After the material had been implanted without BMP-2 for 28 days, the chitosan gel was resorbed, and collagen fibers took its place. Polylactide granules were resorbed by foreign-body giant cells. Bone tissue formed only in the region of the bone margins and was absent in the center. The volume of newly formed bone tissue was 14 ± 6% of the total material volume (Figure 3 and Figure 4).

Implantation of the material with 10 μg/mL of rhBMP-2 led to the bone formation in both the center and the bone defect margin. A newly formed bone surrounded the residues of the polylactide granules. The newly formed bone tissue volume was 56 ± 25% (Figure 3 and Figure 4).

The material extent beyond the bone defect after implantation of the material with 50 μg/mL of rhBMP-2 was observed. The regeneration zone significantly exceeded the size of the defect. Inside the bone regenerate, lacunae containing bone marrow were determined. A dose-related effect of BMP-2 was demonstrated: the higher the concentration of the osteoinductor used, the greater the bone regeneration volume. When implanting a composition with 50 μg/mL of rhBMP-2, the volume of the newly formed bone tissue relative to the volume of material (Nb.Ar.%) was 64 ± 4% (Figure 3 and Figure 4). However, the bone formation was excessive and led to the displacement of the material, and the volume of bone tissue relative to the volume of the original defect exceeded 106%.

The absence of bone formation when using material without BMP-2 and the excessive growth of bone tissue using 50 μg/mL showed that the dosage of 10 μg/mL BMP-2 was desirable: it led to homogeneous newly formed bone distribution within the implanted composition without displacement.

### 2.4. Induction of Ectopic Bone Formation

The osteoinductive potential of the composition based on the chitosan hydrogel with 39 DD% and 12 wt% highly porous polylactide granules impregnated with 10 μg/mL BMP-2 was shown. The composition induced ectopic bone formation when it was implanted subcutaneously (Figure 5). The largest accumulation of newly formed bone tissue was observed on the periphery. In the center of the material, there was a small amount of newly formed bone. It surrounded some polylactide granules and was ingrown into their thickness. The chitosan hydrogel was resorbed and replaced by collagen fibers. These data indicate a high osteoconductive ability of the composition associated with vascular invasion and cell migration. Foreign-body giant cells were detected around the polylactide granules. Plasma cells and lymphocytes were found inside the material. Solitary leukocytes and a few leukocyte clusters were found. This histological pattern corresponded to the resorption process of polylactides [13,14].

## 3. Discussion

A comprehensive in vitro and in vivo study of chitosan/β-glycerophosphate hydrogels showed that a high degree of deacetylation of chitosan led to acute inflammation and cell death. In the literature, some in vitro studies have shown opposite results [16,17,18]. Chitosan samples with a high degree of deacetylation contributed to increasing the hydrophilicity of materials and attaching cells to their surfaces [19,20]. According to these studies, the results were due to nonspecific electrostatic interactions between protonated amino groups of chitosan and the cell membrane surface. In another in vitro experiment on the fibroblast cell line, no effect of deacetylation on chitosan films’ biocompatibility was reported [21]. This could be related to using chitosan samples with different degrees of deacetylation. Various chemical characteristics, including molecular weight and ash, may have significantly impacted the properties of chitosan and chitosan-based materials [17,22,23]. We associated the differences in these results with using in the experiments cell lines other than MSC. Moreover, the form of different chitosan-based materials could vary in several characteristics that impact the properties of the materials, apart from the degree of deacetylation. We previously performed a similar study with an evaluation of the cytocompatibility of different chitosan forms with MSC. There was no significant difference between the cell viability on the different chitosan-based material forms [24]. So the use of a one-stock chitosan sample with different degrees of deacetylation for studying the mechanical and biological properties of chitosan-based materials was an essential feature of this work.

The study results also showed that reducing the degree of deacetylation of the chitosan to 39% still obtained gel-like substances compared with chitosan with a 19.5 DD%. Moreover, it improved the biological properties compared with a composition based on chitosan with a high degree of deacetylation. In vitro use of a composition based on chitosan with a 39 DD% resulted in high cytocompatibility, but leukocyte infiltration was still observed in vivo. The use of highly porous polylactide granules solved this problem. The reduced leukocyte infiltration may have been related to two aspects: firstly, a decrease in the amount of gel in the material’s volume due to its replacement with PLA granules; secondly, the gel’s distribution over the surface of highly porous PLA granules led to an increase in its contact area with interstitial fluid and blood plasma proteins. In addition to improving the material’s biological properties, the polylactide granules also improved the mechanical characteristics of reacetylated chitosan hydrogels. The optimal ability for modeling had gels filled with 12 wt% PLA granules: their modulus of elasticity was about 100 times higher than that of pure gels. Filling to 20 wt% and higher led to disruption of the hydrogel phase continuity and a decreased number of effective crosslinking sites. As a result, destruction of the material at a deformation of more than 10% was shown. Similar data were obtained when studying other systems where the main components were a chitosan hydrogel and particles from polylactide or polylactide-co-glycolide [25,26].

Impregnation of BMP-2 enhanced the osteoinductive properties of the composition based on the reacetylated chitosan/β-glycerophosphate with polylactide granules. The material contained BMP-2 at a concentration of 2–3 orders of magnitude lower compared with the known commercial material Infuse Bone Graft (Medtronic, Minneapolis, MN, USA) with BMP-2 at a concentration of 1.5 mg/mL. However, their effectiveness for bone tissue regeneration was comparable, and in some cases, was exceeded when the Infuse material was used. In one of the studies, by the end of 28 days, the Infuse Bone Graft with 1.5 mg/mL of BMP-2 caused approximately 11.5 ± 1.5% new bone formation in critical-size calvarial defects [27]. This was significantly lower than those in our study. In another study, where Infuse Bone Graft components consisting of a collagen sponge soaked with 2.50 μg/mL of rhBMP-2 were used in the critical-size calvarial defects in rats, the volume of newly formed bone was 99.4 ± 1.8% [28]. In that study, the authors did not note a significant increase in bone tissue volume in the regenerate when a high concentration of BMP-2 was used. Such results could be related to the fact that the authors calculated the volume of the regenerate relative to the volume of the original defect. Moreover, excessive growth of bone tissue beyond the defect was taken into account, which led to an overstatement of the results, bringing them closer to 100%.

In our study, using a concentration of 50 μg/mL in the composition of a chitosan gel with polylactide granules led to more than 106% formation of bone relative to the volume of the bone defect. The volume of the newly formed bone was 64 ± 4% relative to the volume of the implanted material. As a result, the volume of the newly formed bone tissue exceeded the relative volume of the defect and went beyond its limits. This effect can be described as hyperostosis, a dangerous effect of high-dose BMP-2 use. In our study, using a composition with 10 μg/mL of BMP-2 led to the formation of 56 ± 25% bone in the material that was not statistically significantly different from using a composition with 50 μg/mL. In addition, the material with 10 μg/mL of BMP-2 induced ectopic osteogenesis in the absence of a bone environment. The formation of multiple areas of bone tissue in the center of the material indicated a high osteoconductive ability of the material. The porosity of the polylactide and the sufficient rate of gel resorption allowed the ingrowing of vessels, as well as cells to migrate and to spread to the front of newly formed bone tissue [29].

The results of the complex studies indicated that the developed material based on reacetylated chitosan, β-glycerophosphate, and highly porous polylactide granules with BMP-2 was highly effective and could be used in the future for bone tissue regeneration.

## 4. Materials and Methods

### 4.1. Chitosan Hydrogel

Chitosan (cat. 43040, Sigma Aldrich, Saint Louis, MO, USA) was dissolved in 0.1 M of acetic acid to prepare a 2.2% solution for 3 days. At a temperature of 4 °C, a sterile cooled 50% aqueous solution of β-glycerophosphate (Sigma Aldrich, cat. 50020) was added dropwise to the chitosan solution with constant stirring to a final concentration of 20%. Chitosan gels were prepared under aseptic conditions in a laminar flow hood.

### 4.2. Chitosan Reacetylation

The chitosan samples of various DD% were prepared by reacetylation in an aqueous alcoholic medium in a standard way (Figure 6); 1 g of reprecipitated chitosan (65 DD%) was dissolved in a 1% acetic acid solution and methanol 37% *v/v* with stirring for 30 min. Acetic anhydride was used as the acetylating agent. The reaction was carried out in two stages: (1) half of the concentration of acetic anhydride was poured into the flask and the reactions were carried out for 30 min at 22 °C, following which the chitosan solution was titrated with NaOH to pH 7.0; (2) the remaining amount of acetic anhydride in methanol was added, and the reaction was carried out for 3 h. After the solution’s pH was adjusted to 8.0, the solution was dialyzed against distilled water to obtain chitosan with DD% of 19.5, 39.0, 48.8, and 55.3.

The initial degree of deacetylation of the chitosan was confirmed by means of electrometric titration on SevenMulti pH meter S80 (Mettler Toledo, Gieen, Germany). First, 0.2 g of chitosan was dissolved under constant stirring on a magnetic stirrer for 1 h in 20 mL of a 0.1 N HCl solution. The resulting solution was titrated with 0.1 N of NaOH to pH 10.5. A solution of 0.1 N of HCl was prepared from a fixanal in a 1 L measuring flask. To prepare a 0.1 N solution 40 g of NaOH was added in a flask with a 1 L volume. The exact concentration of the resulting solution was established by titrating three samples of 10 mL volume with a 0.1 N HCl solution. As the reacetylation progressed, the solubility of the chitosan decreased. In this regard, the extent of deacetylation established was by means of the Fourier Perkin Elmer Spectrum 100 spectrometer with the prefix of disturbed total internal reflection (FTIR). FTIR spectra of chitosan: 3450 cm^−1^ O–H and N–H stretching, C–H stretching at 2930 and 2870 cm^−1^, and C=O absorption at 1650 cm^−1^ (Figure 7).

### 4.3. Highly Porous Polylactide (PLA) Granules

Highly porous polylactide granules were used as a biocompatible filler of the chitosan hydrogel. Due to the addition of granules, the contact area of chitosan with tissue fluid and blood plasma increased. The PLA granules were prepared by spraying and freeze-drying Nature Works PLLA (4032D) polymer emulsions dissolved in 1,4-dioxane. The PLA granules had a porosity of 98% and a diameter of 1–2 mm. The granules were sterilized by gamma irradiation of 15 kGy.

### 4.4. BMP-2 Impregnation

Stock solutions with 10 μg and 50 μg of BMP-2 were used. Recombinant human BMP-2 (rhBMP-2, AkronBiotech, Boca Raton, FL, USA, SKU: AK8356, derived by Escherichia coli) was dissolved in 100 μL of buffer containing 100 μg of BSA and 20 μL of 10 mM acetic acid. The PLA granules were wetted by a rhBMP-2 solution and vacuumed five times. Subsequent freeze-drying led to protein adsorption on the surface and in the pores of the polylactide granules.

### 4.5. Chitosan Hydrogel Filled with PLA Granules

Sterile high-porosity polylactide granules were added to the cooled chitosan solution. The concentration of the PLA granules was 1–20 wt%. The chitosan hydrogel and PLA granules were mixed with a spatula until homogenous distribution. At this stage, the gel saturate granules and the viscoelastic properties of the composition increased.

For the in vitro studies, the composition was packaged in wells of a 96-well culture plate and incubated on a heated platform for 2 h at +37–40 °C until cured.

For the in vivo studies, compositions of 0.125–0.25 mL were packaged in insulin syringes without a cannula.

### 4.6. Material Mechanical Testing

The mechanical strength of the composite material was analyzed by measuring the tensile strength and Young’s Modulus at room temperature using a universal testing machine (UTM) (Instron 5583, Instron Corp., Norwood, MA, USA). The deformation speed was 50%/min with a cylindrical sample size of 10 mm width and ~10 mm length. Compression tests were performed using a universal testing machine (Instron 5965, Instron Corp., Norwood, MA, USA).

### 4.7. In Vitro Studies

Human adipose tissue-derived MSCs were used. The MSCs were cultured in a medium consisting of Dulbecco’s Modified Eagle Medium (DMEM; PanEco, Moscow, Russia) with 10% fetal bovine serum (FBS; PAA Laboratories Inc., Toronto, Canada), 0.584 mg/mL of L-glutamine (PanEco, Moscow, Russia), 10 ng/mL of recombinant human Fibroblast Growth Factor type 2 (rhFGF-2; ProSpec Tany, Israel), 5000 U/mL of heparin sodium (PanEco, Moscow, Russia), 5000 U/mL of penicillin (PanEco, Moscow, Russia), and 5000 µg/mL of streptomycin (PanEco, Moscow, Russia) at 37 °C and 5% CO_2_.

Cell viability was assessed by the MTT assay. Cells were incubated with 0.5 mg/mL 3-(4,5-dimethylthiazol-2-yl)-2,5-diphenyl tetrazolium bromide (MTT; PanEco, Moscow, Russia) for 2 h at 37 °C. Then the crystals of formazan were eluted using dimethyl sulfoxide (DMSO; PanEco, Moscow, Russia) and the formazan absorption was measured on xMark (Bio-Rad, Hercules, CA, USA) at a wavelength of 570 nm, subtracting the background value at 620 nm.

To study the cell adhesion of the chitosan-based hydrogels, MSCs were stained using a PKH26 dye (Sigma Aldrich, Saint Louis, MO, USA) according to the manufacturer’s recommendations.

### 4.8. In Vivo Studies

This experiment used 28 male Wistar rats weighing 300–350 g. The rats were randomly divided into four groups depending on the type of the material and implantation site. Each group included seven rats. The biocompatibility was tested in a subcutaneous model. Osteoinductive properties were evaluated on a critical-size calvarial defect model. All protocols in this study were approved by the Committee on the Ethics of Animal Experiments of Central Research Institute of Dentistry and Maxillo-facial Surgery, Moscow, Russia (IACUC permit no:2019-293), in compliance with the Guide for the Care and Use of Laboratory Animals published by the U.S. National Institutes of Health (NIH publication no. 85-23, revised 1996), the European Convention for the Protection of Vertebrate Animals used for Experimental and Other Scientific Purposes, and ISO 10993-2, 2006. The rats were intraperitoneally anesthetized with 30 mg/kg of Zoletil (Virbac, Carros, France) and 5 mg/kg of Xylazine (Interchemie Werken De Adelaar B.V., Venray, Netherlands).

Subcutaneous implantation: One dorsum midline incision was made after shaving and ethanol disinfection of the outer skin. Two pockets on each side of the incision were made. The materials were implanted into subcutaneous pockets, and the wounds were sutured with Vicryl 5/0 (Ethicon, Somerville, NJ, USA).

Implantation into critical-size calvarial defects: Operations were performed following the previously developed method [12]. After shaving and disinfection, transverse and vertical laterally displaced incisions of the scalp were made, forming a triangular flap. The parietal bones were subsequently exposed bluntly and sharply. A full-thickness calvarial bone defect was created without venous sinus perforation using a C-reamer trepan of a 5.5 mm diameter and a 1.5 mm height from the SLA Kit (Neobiotech, Seoul, Korea). After implantation, the periosteum and skin were closed with 5.0 Vicryl sutures.

Euthanasia: Rats were euthanized by CO_2_ inhalation after 28 days. The calvarias were submitted to histological study.

Histological examination and morphometry: Calvarias were set in 10% neutral formalin for 24–48 h, washed in running water for 24 h, decalcified in 20% EDTA for 5 weeks, dehydrated in alcoholic solutions, and paraffin was embedded. The thickness of the tissue sections was 5–7 μm. Slides were stained with hematoxylin–eosin (H&E) and Masson trichrome (with aniline blue) (BioVitrum, Saint-Petersburg, Russia). The slides were scanned on an Axioimager M.1 light microscope (Carl Zeiss, Göttingen, Germany) using the Zen Pro 3.0 software complex. Morphometric analysis was performed on the six serial sections for each sample. The volume of newly formed bone was evaluated. Morphometric analysis was made in accordance with generally accepted recommendations [13,14].

### 4.9. Statistical Analysis

Statistical analysis and graphing were performed with GraphPad Prism 7.0 (USA). Intergroup differences were statistically analyzed by a Student’s *t*-test. Differences were considered significant when *p* < 0.05.

## Figures and Tables

**Figure 1 molecules-26-00261-f001:**
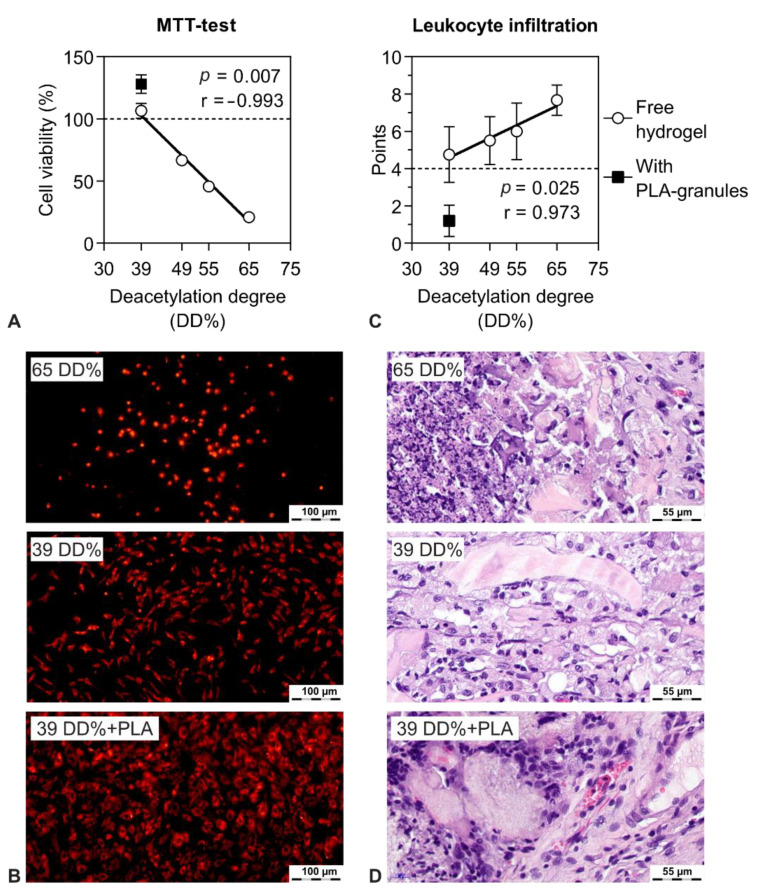
The effect of chitosan reacetylation on the biocompatibility of the thermosetting chitosan–β-glycerophosphate hydrogel. (**A**) MSC viability in the presence of chitosan hydrogels, seven days, MTT test. (**B**) Cell adhesion of the chitosan-based hydrogels at the lowest and highest degrees of deacetylation (65 and 39 DD%), PKH26 staining. (**C**) Level of leukocyte infiltration 14 days after subcutaneous implantation of chitosan hydrogels. (**D**) Tissue sections of the implantation area of the chitosan-based hydrogels with high and low degrees of deacetylation (65 and 39 DD%).

**Figure 2 molecules-26-00261-f002:**
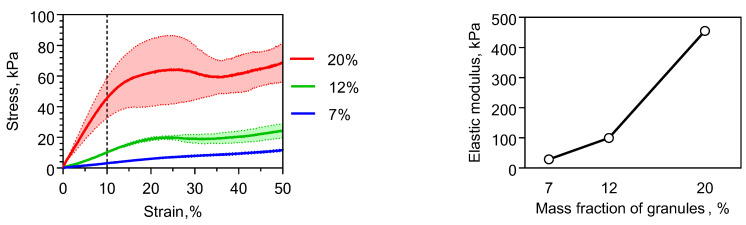
Mechanical properties of composition based on the chitosan and β-glycerophosphate hydrogel and the PLA granules. (**Left**) The deformation curves of the composite materials during compression test. (**Right**) The elastic modulus of the compositions containing different PLA granule mass fractions.

**Figure 3 molecules-26-00261-f003:**
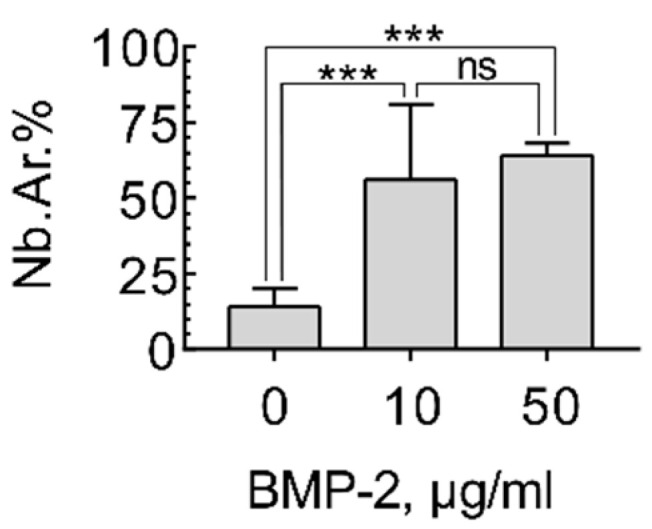
Volume of newly formed bone in the critical-size calvarial bone defect after implantation of the composition based on the reacetylated chitosan/β-glycerophosphate hydrogel and 12 wt% PLA granules containing different BMP-2 doses in rats after 28 days. The significance of differences was indicated in accordance with the requirements of the American Psychological Association (APA): ns—*p* > 0.05; ***—*p* ≤ 0.001.

**Figure 4 molecules-26-00261-f004:**
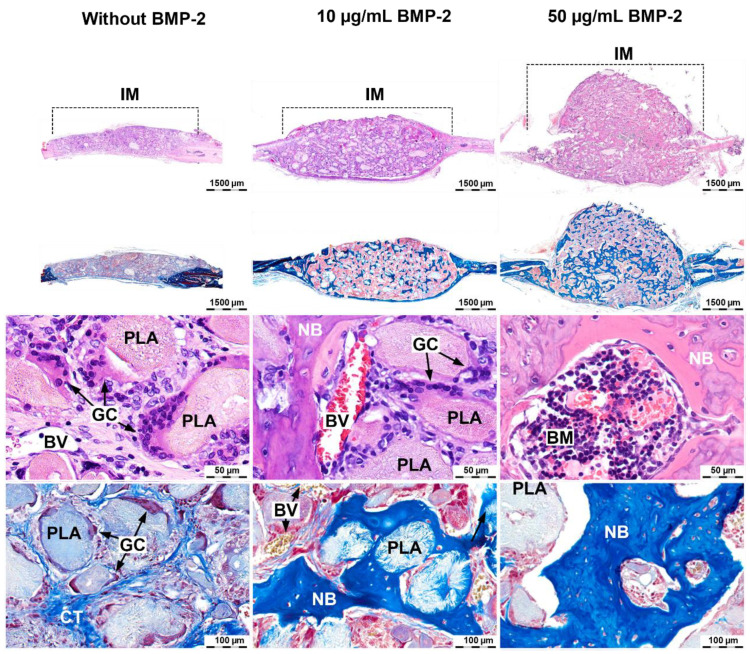
Bone regeneration capacity in the critical-size calvarial bone defect area of rats after implantation of the material based on the reacetylated chitosan/β-glycerophosphate hydrogel and 12 wt% PLA granules containing 0, 10, and 50 µg/mL of BMP-2. H&E and Masson’s trichrome staining after 28 days. PLA—highly porous polylactide granules, NB—newly formed bone, BV—blood vessel, GC—foreign-body giant cell, CT—connective tissue, IM—implanted material, and BM—bone marrow.

**Figure 5 molecules-26-00261-f005:**
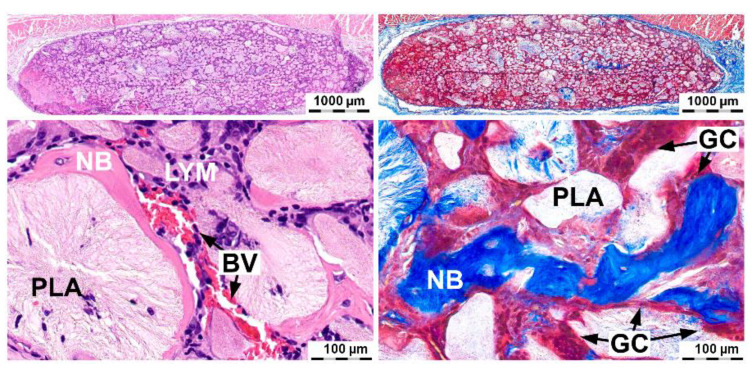
Bone formation after subcutaneous implantation of the composition containing 10 µg/mL of BMP-2. H&E and Masson’s trichrome staining after 28 days. PLA—highly porous polylactide granules, NB—newly formed bone, BV—blood vessel, GC—foreign-body giant cell, LYM—lymphocytes, and plasma cells.

**Figure 6 molecules-26-00261-f006:**
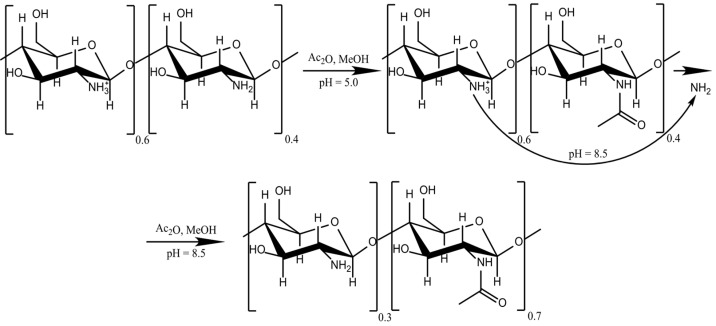
Scheme of a two-step chitosan acetylation reaction.

**Figure 7 molecules-26-00261-f007:**
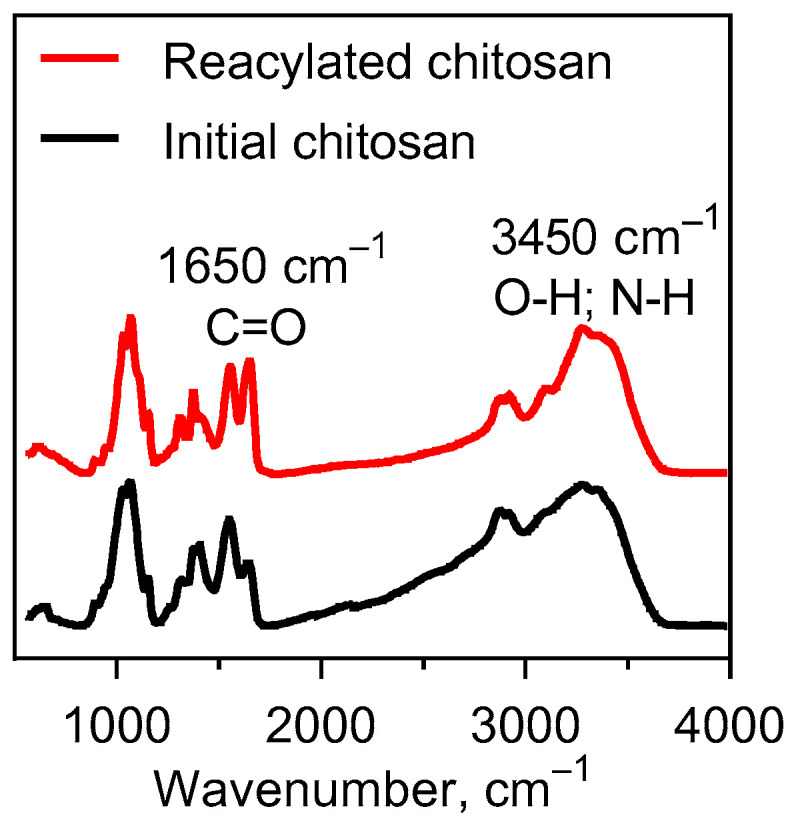
Infrared spectroscopy for the initial and reacylated chitosan.

## Data Availability

Available upon request.

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
