# Peer review of "Influence of the Degree of Deacetylation of Chitosan and BMP-2 Concentration on Biocompatibility and Osteogenic Properties of BMP-2/PLA Granule-Loaded Chitosan/β-Glycerophosphate Hydrogels"

_molecules, 2021, doi:10.3390/molecules26020261_

Round 1
Reviewer 1 Report
This paper provides good and interesting results with some novelty aspects. It is well organized and fairly well described
There are a major remark and a minor one
Major remark: the legends of the figures are very concise and not very explanatory. Please provide a better explanation, especially for Figures 2 and 5. Also note that at point 2.2, the finding:"Excessive filling of hydrogel with PLA granules more than 20% leads to material destruction at ~10% strain" is not shown in Figure 2. Therefore, mentioning Figure 2 there is incorrect
Minor remark: Under point 2.1 two Figures not existing are mentioned (i.e., Figure 3b and Figure 11)
Author Response
Thank you very much for your review!
***
We add more information into 2 and 5 figures caption. Figure 2 shows the deformation curves of the composite materials during the compression test and their elastic modulus. Figure 5 shows bone formation after subcutaneous implantation of the composition containing 10 µg/ml BMP-2 in H&E and Masson's trichrome staining.
In Figure 2 we correct the dotted line position, and information in the text corresponds to graph information.
***
We fix errors and write the correct numbers of figures.
***
Detailed information is provided in the attached file, where the review function shows which points have been corrected.
Reviewer 2 Report
The authors proposed a composite hydrogel based on chitosan/β-glycerophosphate with different degrees of deacetylation, in order to promote the osteoinductive and osteoconductive properties. The paper is well written in English. Figures should be revised according to the suggestion below.
- Figure 1 shows the results for MTT test and Leukocytes infiltration. In both graphs, it is not clear the data regarding the PLA-granules samples (the squared ones). It seems that there is only one data along the time of the experiment. The authors should clarify this point.
- The figure 4 should have the images at the same magnification (referring to the first two rows). This would give a better understanding of the details of the images.
- The figure 5 could be improved by adding a better figure caption that can explain the type of staining performed and what the images want to show.
Author Response
Thank you very much for your review!
***
Figure 1
PLA granules were presented in only one composition with reacetylated chitosan gel. This information is indicated in the results and the materials and methods sections. We added data to illustrate the difference in cyto- and biocompatibility between reacetylated 39 DD% chitosan gel and the same gel with PLA granules and show that PLA granules addition improves biological properties of chitosan hydrogel.
Figure 4
We change images to that with the same magnification.
Figure 5
We correct the description and add information about staining in Figure 5 caption.
***
Detailed information is provided in the attached file, where the review function shows which points have been corrected.